

# Cloud-edge MQTT messaging for latency mitigation and broker memory footprint reduction

Yi-Hsuan Tseng, Chao Wang, Yu-Tse Wei and Yu-Ting Chiang

Department of Computer Science and Information Engineering, National Taiwan Normal University, Taipei City, Taiwan

## ABSTRACT

The deployment of smart-city applications has increased the number of Internet of Things (IoT) devices connected to a network cloud. Thanks to its flexibility in matching data publishers and subscribers, broker-based data communication could be a solution for such IoT data delivery, and MQTT is one of the widely used messaging protocols in this class. While MQTT by default does not differentiate message flows by size, it is observed that transient local network congestion may cause size-dependent latency additions, and that the accumulation of large message copies in the cloud broker could run out of the broker memory. In response, in the scope of cloud-edge messaging, this research article presents problem analysis, system design and implementation, and empirical and analytical performance evaluation. The article introduces three message scheduling policies for subscribers deployed at network edge, and a memory allocation scheme for MQTT broker deployed at network cloud. The proposed design has been implemented based on Eclipse Mosquitto, an open-source MQTT broker implementation. Empirical and analytical validations have demonstrated the performance of the proposed design in latency mitigation, and the result also shows that, empirically, the proposed design may save the run-time broker memory footprint by about 75%. Applicability of the proposed design to other messaging services are discussed by the end of the article.

# INTRODUCTION

Internet of Things (IoT) technology has helped modern society and its administration to collect data and orchestrate city-scale smart applications (*Bellini, Nesi & Pantaleo, 2022*; *Gharaibeh et al., 2017*). The types of data collected range from small-size sensing data (*Ullah et al., 2024*; *Allen & Melgar, 2019*) to large-size video data (*Pathak et al., 2024*; *Tian et al., 2018*). In many such applications, besides offline data analysis, online data processing is critical and requires a suitable data delivery network protocol.

Data delivery in smart-city applications may travel a long route, as the software ecosystem is often deployed across a network cloud and local networks that are at the edge of the cloud. Deployed at the city scale, at one end, data publishers are IoT data-collecting devices such as sensors and cameras; at the other end, data subscribers are applications that may be deployed either in the network cloud or at the edges of the cloud. Between

Corresponding author
Chao Wang, cw@ntnu.edu.tw

publishers and subscribers, a software service called broker is deployed in the network cloud and helps forward data while providing the needed quality-of-service (QoS) levels.

Among network data delivery protocols (*Naik, 2017*), MQTT (*Banks et al., 2019*) is flexible in matching data publishers and subscribers and has inherent support of the QoS for data delivery guarantees. For example, MQTT can be used to implement the idea of *geofence* (*Rodriguez Garzon & Deva, 2014*) at the city scale for location-aware servicing: by associating topics for publish/subscribe to geographic regions, clients subscribing to such a topic may receive data relevant to specific locations. One immediate benefit is messaging filtering and data traffic reduction.

But MQTT also adds peculiar challenges to the cloud-edge data communication. In MQTT, as all message deliveries have to go through the broker, the cloud broker may become a bottleneck of both temporal performance (data delivery latency) and spatial performance (memory and storage demand at the broker). Consider a generic data traffic pattern in smart-city applications: small-size messages targeting at low latency (*e.g.*, event notifications) and large-size messages targeting at high reliability (*e.g.*, detail account of an event), and that each message published may need to be delivered to multiple subscribers. Using MQTT, in its attempts to deliver a large-size message downstream a temporarily congested edge network, the cloud broker could delay other messages; moreover, the large-message copies queued up within the broker could quickly consume the memory on the broker host. This could be a problem in particular when the broker is deployed in a cloud VM with limited memory capacity.

Accordingly, this article describes a study on the following research question: *Given different-sized message streams in cloud-edge MQTT messaging scenarios, and with transient network congestion, how to both limit latency penalty on each message and bound memory consumption on the broker host?* The research contributions of this work include the following items:

1) *A problem analysis and empirical performance micro-benchmark*: This article presents two experiments that demonstrate both the temporal and spatial performance issues in cloud-edge messaging, as mentioned above. Multiple pairs of data communication were conducted across the network cloud and edge networks, with an MQTT broker deployed in an AWS EC2 VM.

2) *A set of message-size-aware scheduling policies for an MQTT broker*: three adaptive message scheduling policies are proposed to reduce the small-size message latency while bounding the large-size message latency. The scheduling policies take into account downstream network conditions and withhold large-size message deliveries in favor of small-size message latency performance. The decision to resume a large-size message delivery is based on historical and estimated network conditions.

3) *An improved memory allocation strategy for a cloud MQTT broker*: a lazy memory allocation strategy is proposed to decouple the relation between the size of run-time memory footprint and the number of subscribers of large-size messages.

4) *An implementation of the proposed design*: the proposed design has been implemented based on an open-source MQTT broker named Eclipse Mosquitto

(https://mosquitto.org/), and the implementation of the design is available as open source software (https://github.com/wangc86/Adaptive-MQTT-Transmit-Policy).

5) *Empirical and analytical performance evaluation*: the implementation has been evaluated in a cloud-edge deployment setting, and comprehensive analytical analysis is conducted to show the performance nuances of the proposed design under a range of design parameters.

The rest of this article is structured as follows. "Related Work" surveys known results in general network performance management, related work in integrating MQTT with some other protocols, and the ones that are specific for MQTT. "System Model and Problem Analysis" describes the system model and problem analysis, with two micro-benchmark experiments motivating the research question and solution. "Cloud MQTT Broker Design" introduces both the scheduling policies and the memory allocation strategy for MQTT cloud broker. "Implementation and Performance Validation" describes the implementation and performance evaluation, with empirical results and analytical analyses. Subsequently, "Discussion" gives a list of configuration recommendations for the usage of the proposed work, and a discussion is included on how the experimental parameters affect the empirical results, and how the proposed approach generalizes to different MQTT brokers and other broker-based systems. "Concluding Remarks" concludes the article.

## RELATED WORK

Fundamentals in how data routing and flow control impact data network latency have been known since the early days of computer networks research and development (*Bertsekas & Gallager, 1992*), and queueing theory has shown itself applicable in many practical settings (*Harchol-Balter, 2013*). Nevertheless, given the increasing capacity of the modern network core in both bandwidth and speed, and as new applications have become more tangible (such as those in Tactile Internet; *Promwongsa et al., 2020*), both network resource management (*Shakarami et al., 2022*) and quality-of-service guarantee (*Zolfaghari et al., 2020*) have become yet more important. In contrast to clean-slate designs that target at the network core (*Anjum et al., 2024*), MQTT takes the approach at the application layer and is agnostic to the network functionalities from the layers beneath.

A comparative study on MQTT and HTTP (*Yokotani & Sasaki, 2016*) shows that the topic length of a MQTT message could significantly impact network latency and therefore suggested a solution for topic compression. An analysis on message loss and delay in MQTT (*Lee et al., 2013*) shows that message loss under various payloads is correlated with end-to-end delay. To better handle heterogeneous data size, a solution is to use both MQTT and FTP (*Ohno et al., 2022*), where MQTT handles small data and FTP handles large data. A more recent work (*Li, Chiang & Wang, 2024*) compares broker-based and broker-less data deliveries, and an adaptive scheme is introduced for switching traffic between the two for better video streaming quality.

Many studies have recognized the challenges posed by placing MQTT brokers in the cloud, and some resorted to the idea of deploying brokers at the edge to share the load of

the cloud broker (*Banno et al., 2017*). In the work LA-MQTT (*Montori et al., 2022*), an approach leverages collaboration among multiple brokers and use geographical hints to find the nearest broker for data transmission. As deploying brokers at the edge could lead to challenges in managing a large number of brokers, researchers (*Park, Kim & Kim, 2018*) proposed to use software-defined networking (SDN) with MQTT brokers to set up better data delivery paths. In the case of a pool of subscribers, A related study (*Matić et al., 2020*) described a method to assess the load of each subscriber and accordingly determine which subscriber to send the data to. The work EMMA (*Rausch, Nastic & Dustdar, 2018*) proposed to use network latency to infer the distance between clients and brokers and redirect clients to their closest broker. A more recent work, MQTT-SD (*Kamoun et al., 2024*), proposed to use data fusion and aggregation to reduce the volume of data flow to subscribers, and the approach can be integrated into existing MQTT infrastructure.

## SYSTEM MODEL AND PROBLEM ANALYSIS

In MQTT, data communication takes place in terms of messaging between publishers and subscribers. A messaging broker matches publishers and subscribers if they share the same message topic. To meet the requirement of different applications, MQTT defines three Quality of Service (QoS) levels for message delivery: QoS 0 (at most once delivery), QoS 1 (at least once delivery), and QoS 2 (exactly once delivery). QoS 2 is rarely used; hence, this study focuses on QoS 0 and QoS 1. If there are multiple subscribers subscribing to the same topic, the broker would make individual delivery attempt for each subscriber.

Figure 1 illustrates a generic MQTT messaging scenario that span across the network cloud and multiple local networks at the edge of the cloud. If both that the subscribers are many and that each message is large (*e.g.*, the yellow message flow), then the broker could add much traffic load to the downstream network. Now, if the downstream network is temporarily congested (red areas in Fig. 1), then packets could be dropped and future delivery attempts may further congest that part of the network and affect the other message flows (*e.g.*, the green message flow in edge network 3). Furthermore, under such condition the broker will have to keep messages for future delivery attempts, and this could quickly consume the memory in the broker. The following two experiments demonstrate the problem.

In the first experiment, a Mosquitto MQTT broker (*Light, 2017*) (version 2.0.12) ran in a VM in Amazon EC2 us-east region, and 10 publishers and 100 subscribers ran on local PCs in Taiwan. The broker VM had 1 GB memory. The detail configurations are listed in Tables 1 and 2. Figure 2 shows the results of latency from publisher to subscriber. As Fig. 2A illustrates, the latency is not much influenced by the QoS level as is by the message size, possibly because the PUBACK control packet for QoS 1 messaging acknowledgment is small packet (https://mqtt.org/mqtt-specification/). Furthermore, large-size messages are more sensitive to changing network condition (Fig. 2B), as further latency breakdown shows that the average message processing time in the broker is less than 1% of the overall end-to-end latency, whereas the time of publisher-to-broker is about 46.6%, and the time of broker-to-subscriber is about 52.4%.

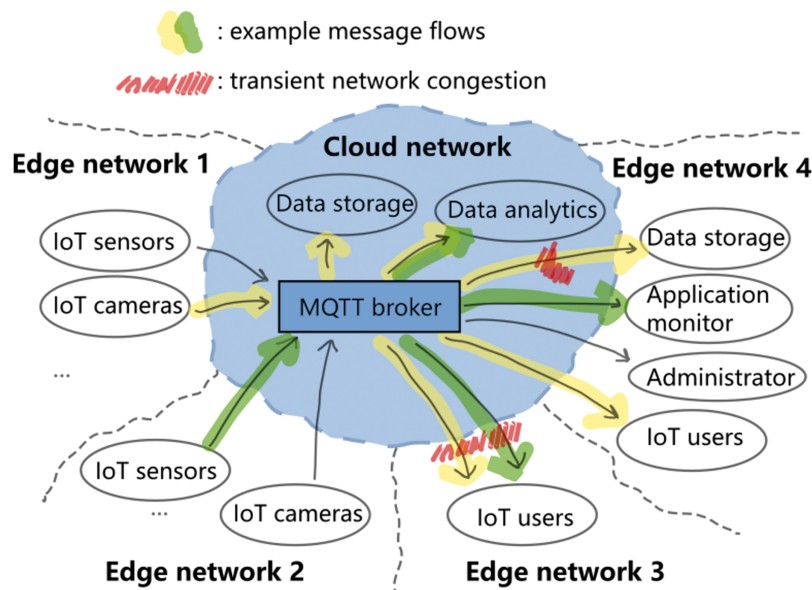

**Figure 1 Generic cloud-edge MQTT messaging scenario.**

**Table 1 Publishers setting of benchmark experiment.**

| Publisher ID | QoS | Payload size | Sending period | Topic | Subscribers |
|---|---|---|---|---|---|
| PUB1 | 1 | 100 B | 10 s | QoS1/10 B | 46 |
| PUB01 | 0 | 100 B | 10 s | QoS0/10 B | 46 |
| PUB2 | 1 | 1 KB | 10 s | QoS1/1 KB | 46 |
| PUB02 | 0 | 1 KB | 10 s | QoS0/1 KB | 46 |
| PUB3 | 1 | 10 KB | 10 s | QoS1/10 KB | 46 |
| PUB03 | 0 | 10 KB | 10 s | QoS0/10 KB | 46 |
| PUB4 | 1 | 100 KB | 10 s | QoS1/100 KB | 46 |
| PUB04 | 0 | 100 KB | 10 s | QoS0/100 KB | 46 |
| PUB5 | 1 | 1 MB | 10 s | QoS1/1 MB | 46 |
| PUB05 | 0 | 1 KB | 10 s | QoS0/1 MB | 46 |

**Table 2 Subscribers setting of benchmark experiment.**

| Subscriber ID | QoS | Topic |
|---|---|---|
| SUB1 | 0 | QoS0/100 B |
| SUB2 | 0 | QoS0/1 KB |
| SUB3 | 0 | QoS0/10 KB |
| SUB4 | 0 | QoS0/100 KB |
| SUB5 | 0 | QoS0/1 MB |
| SUB6 | 0 | QoS1/100 B |
| SUB7 | 1 | QoS1/1 KB |

(Continued)

| Table 2 (continued) | | |
| --- | --- | --- |
| **Subscriber ID** | **QoS** | **Topic** |
| SUB8 | 1 | QoS1/10 KB |
| SUB9 | 1 | QoS1/100 KB |
| SUB10 | 1 | QoS1/1 MB |
| SUB11 to SUB55 | 0 | QoS0/# |
| SUB56 to SUB100 | 1 | QoS1/# |

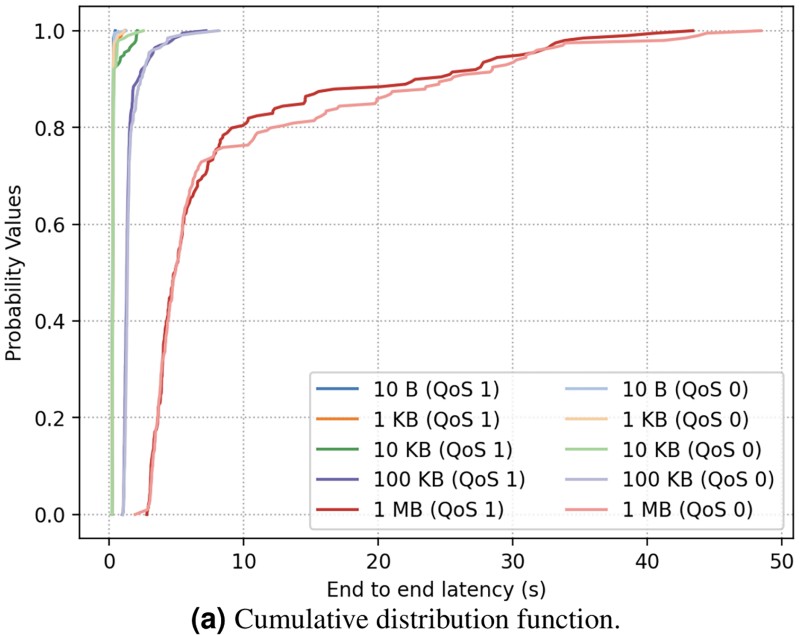

**(a)** Cumulative distribution function.

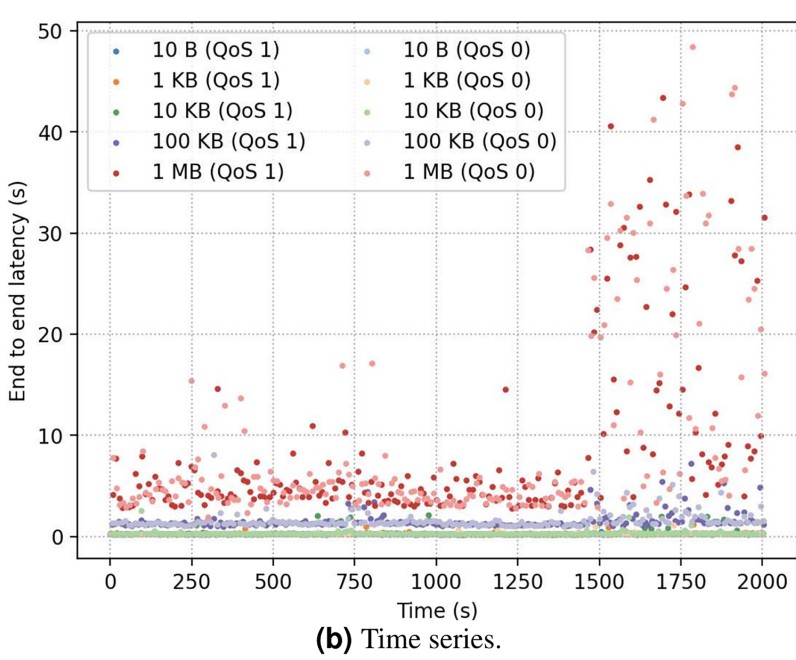

**(b)** Time series.

**Figure 2 (A and B) End-to-end latency under different QoS.**

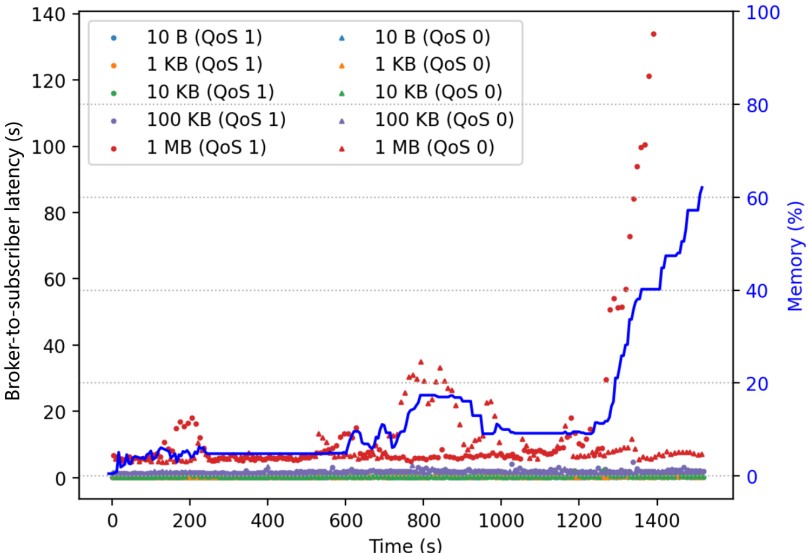

**Figure 3 Correlation between broker memory usage and broker-to-subscriber latency.**

In the experiment, while about 25% of 1 MB of messages were affected by transient network congestion and had much longer latency, about 10% of 100 KB messages were affected, and still lower percentage of smaller messages were affected. This indicates that the cause of additional latency for small-size messages is not the same as that for large-size messages. It is conjectured that smaller messages were impacted more by larger messages going to the same subscribing host than by the vanilla network condition with no messaging.

The second experiment had the same configuration as the first, except that it was conducted during a peak hour of campus network activities. Figure 3 shows the results. Each point in the figure marks the latency of message delivery from the broker to a subscriber, and the blue line shows the memory usage of the broker. The result shows a correlation between messaging latency and the broker memory usage. In particular, when handling the 1 MB packets with QoS 1, the latency rise was accompanied by a significant increase in memory usage. Should the demand of QoS 1 large messages be many, the broker could run out of memory.

Overall, there are four observations from the above two experiments:

1) Messaging latency is subject to both the message size and the network condition and is less to do with the QoS level (either 0 or 1).

2) Large-size messages may suffer from much longer latency penalties under network congestion.

3) Large-size messages may impact the latency of small-size messages should they go to the same subscribing host.

4) Large-size not-yet-delivered message copies could run out of broker memory.

# CLOUD MQTT BROKER DESIGN

This section describes the proposed improvements to MQTT broker design. The motivation is to reduce small-size message latency while bounding large-size message latency. Following the observations from the micro-benchmark experiments (presented in the previous section), the key design idea is to temporarily withhold transmission of large messages until either the downstream network congestion has been improved or the message has been withheld for too long. The key design challenge is to resume transmission at some appropriate timing, so as to improve the overall messaging latency performance while not running out of broker memory. This is achieved by some heuristics taking into account network condition, design parameter, and the withhold duration so far. Figure 4 illustrates the architecture of the corresponding broker design. Upon each message arrival, the broker will label it as either large message or small message. Large messages may be temporarily withheld from delivery. A latency detector fathoms the downstream network latency.

There are four design elements in the proposed design: (A) message size classification, (B) network latency monitoring, (C) large message scheduling, (D) lazy memory allocation. The following subsections describe each design element.

## Message size classification

Message classification is done by comparing the size of the arriving message with a configurable threshold value, named `threshold_s`[1]. The value `Threshold_s` specifies the size above which a message should be considered as a large message. The specific value is determined according to the available network bandwidth in each environment of service deployment. When the broker receives a PUBLISH control message from the publisher, if the payload size of the control message is greater than `threshold_s`, the message will be labeled as a large message, otherwise as a small message. The large message will be kept in a temporary buffer and will be scheduled for transmission according to the selected scheduling policy ("Large Message Scheduling").

## Network latency monitoring

The downstream network condition at each subscriber could be different and time-varying. In the proposed design, the broker will monitor the downstream network condition per message subscriber. The monitoring is carried out by having the broker periodically send each message subscriber a query message using QoS 1. The broker then measures the round-trip time (RTT) based on the PUBACK control packet sent back from the subscriber. The RTT is used for estimating the latency a message might experience in the near future, and the estimation is used for message-withholding decisions. To obtain a better estimation, the query message size could be set to the actual message size for application data delivery; alternatively, to reduce its consumption of the network bandwidth, one could set the query message size to some smaller value and use network profiling to infer the latency estimation for each message size (*e.g.*, those results in Figs. 2 and 3).

---

[1] *s* stands for size.

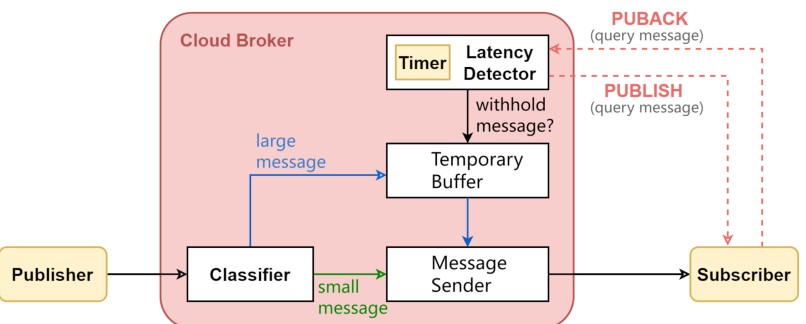

**Figure 4 The proposed MQTT broker design.**

## Large message scheduling

For a large message, the broker may choose to temporarily withhold its delivery, so that (1) it will not bump into the temporarily-congested network and experience significant latency, and (2) it will not further congest the network and delay small-size messages. The broker makes withhold decisions based on a configurable, per-subscriber parameter `threshold_l`[2]. The `threshold_l` represents the RTT under a non-congested network, and it can be either set up as a fixed value or determined at run-time using the moving average of the recent RTTs.

The following notations are used for describing both our design of message scheduling and the performance validation in "Analytical Performance of All Policies". For each message, its duration of withholding thus far is called the *sojourn time* of the message and is denoted by $\Delta_S$. A time length $D$ is as the upper bound of withholding duration. The scheduling decision is updated once $\Delta_S > D$ or once a new latency estimation is available. Let $L_{now}$ be the expected downstream one-way network latency, which is defined to be the latency a message may experience should the broker schedule it now. $L_{now}$ is computed and updated by the following equation:

$$L_{now} = RTT_{latest}/2, \tag{1}$$

where $RTT_{latest}$ is the latest RTT estimation obtained by network latency monitoring. Finally, $T_Q$ is denoted as the sending period of RTT query messages.

This article introduces the following three scheduling policies for large messages:

1. *Conservative policy*: Deliver a message if

$$L_{now} < \texttt{threshold\_l}/2 \ \text{ or } \ \Delta_S > \texttt{D}. \tag{2}$$

Under this policy, put it in another way by plugging in Eq. (1), it says that if $RTT_{latest} > \texttt{threshold\_l}$, the broker would withhold the current delivery attempt unless $\Delta_S > D$ (*i.e.*, when the message has been withheld for too long). The choice of `threshold_l` value needs to be based on the target network environment.

2. *Probabilistic policy*: Deliver a message if

---

[2] *l* stands for latency.

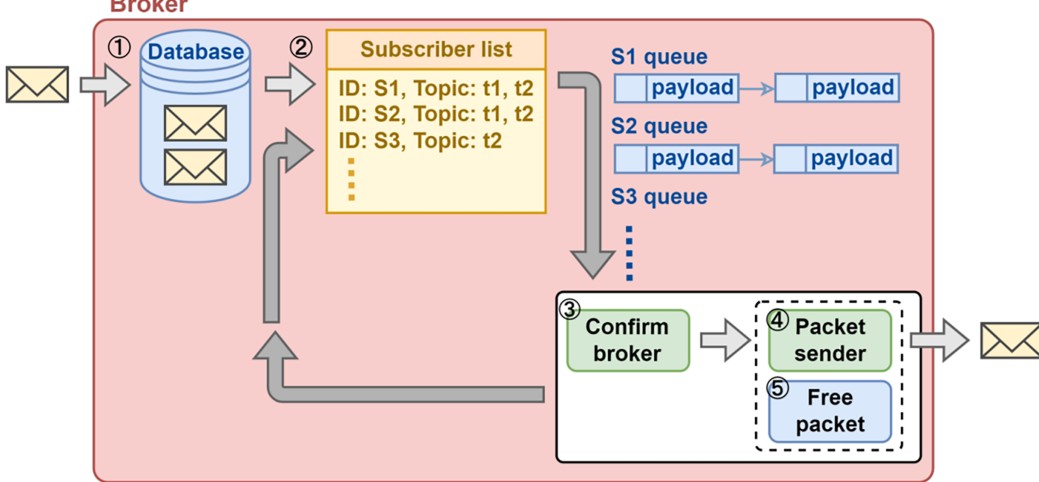

**(a)** Normal memory allocation for messages.

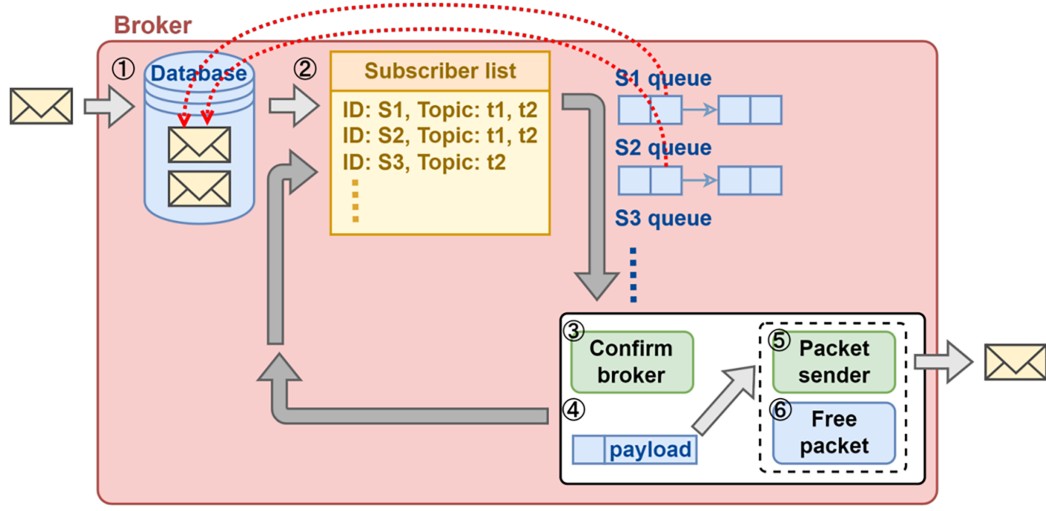

**(b)** Lazy memory allocation for messages.

**Figure 5 (A and B) Broker memory allocation design.**

$$L_{\text{now}} < T_Q + L_{\text{next}} \quad \text{or} \quad \Delta_S > D, \tag{3}$$

where $L_{\text{next}}$ is the expected network latency should the broker send out the message at the next time. The rationale in the probabilistic policy is that we compare the cost of sending message right now *vs* the cost of sending message at the next decision time. The cost is in terms of latency. The value of $L_{\text{next}}$ is determined by the following equation:

$$L_{\text{next}} = (p)(L_{\text{now}}) + (1 - p)(\texttt{threshold\_l}/2). \tag{4}$$

where $p$ is the probability that the network congestion persists at the next decision time. Assuming temporal locality of network congestion, $p$ may be estimated by an exponential distribution (*Harchol-Balter, 2013*):

$$p = e^{-\lambda t}, \tag{5}$$

where $t = \Delta_S + T_Q$ and $\lambda$ is the inverse of average network congestion duration.

3. *Historic policy*: Deliver a message if

$$L_{\text{now}} < T_Q + L_{\text{next}} + \Delta_S, \tag{6}$$

and here for simplicity let $L_{\text{next}} = \texttt{threshold\_1}/2$. Comparing to Eq. (3), the historic policy uses $\Delta_S$ the message sojourn time to cap the withhold time. Here a time limit $D$ is not needed, for the increasing $\Delta_S$ will quickly force the broker to send out the message.

Note that if the RTT query message is of the size different from that of the data message, then both $L_{\text{now}}$ and `threshold_1` need to scale accordingly (*e.g.*, using latency profiling results such as those in Fig. 3). Performance comparison and analysis of these policies will be presented in "Analytical Performance of All Policies".

**Lazy memory allocation**

As illustrated in Fig. 3, even if the broker does not proactively withhold large messages, severe network congestion could still quickly consume the memory resource in the broker. The cause of excessive memory consumption is due to that the broker maintains one sending queue per subscriber and that it allocates memory for each message pushed in the queue (Fig. 5A). This will be a problem for a cloud broker running in a VM with hard limit on memory size when it is serving many subscribers with large messages.

A solution proposal is illustrated in Fig. 5B. Instead of allocating memory for each message subscriber upon queuing operation, the broker pushes a reference to the message kept in the database. The broker makes additional memory allocation only when it is determined that the message can be sent out immediately. This solution de-couples the memory usage and the total number of message subscribers, and therefore it scales well with the increasing number of message subscribers.

## IMPLEMENTATION AND PERFORMANCE VALIDATION

The proposed broker design has been implemented based on Mosquitto version 2.0.12. The RTT measurement was implemented by having a co-host publisher publishing the query messages. The fixed `threshold_1` value was sent by each subscriber in the CONNECT control message while it was establishing a connection to the broker. The rest of this section presents both empirical and analytical performance validations: (1) A set of empirical cloud experiments showing the latency performance of the conservative scheduling policy and the memory usage reduction, and (2) a comprehensive analytical results and analysis on the conservative policy, the probabilistic policy, and the historic policy, against a baseline policy that does not withhold large messages, and a clairvoyant policy that shows the theoretical performance upper bound.

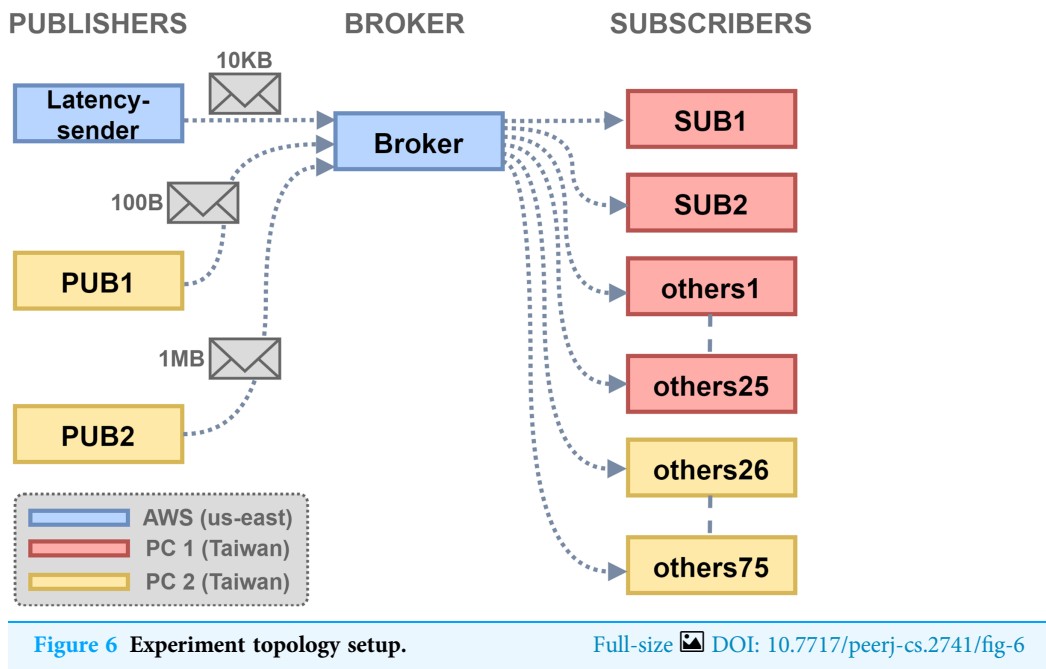

**Figure 6 Experiment topology setup.** 

## Empirical performance of the conservative policy

The experiment topology is shown in Fig. 6. There were three hosts, PC1 and PC2 in Taiwan and a VM instance on the AWS EC2 us-east. The VM that ran the broker has 1 GB memory. The setup simulates an application scenario where MQTT clients are geographically distributed and connections to the broker traverse long-distance network paths. The size threshold `threshold_s` was set to 10 KB, and the RTT query period $T_Q$ was set to 5 s. The latency threshold `threshold_l` was set to 0.5 s, as the empirical one-way latency from edge to cloud was about 0.25 s. The client configuration is shown in Table 3, with two publishers and 77 subscribers in total. SUB1 subscribed to the topic for RTT query, which represents subscriber under the proposed scheduling policy. SUB2 was treated by the default first-come-first-serve broker message forwarding policy, and the remaining 75 subscribers were treated likewise. The Linux `tc` command was set to control the inbound traffic to PC1 to add 100 ms delay and 20% packet drops, to simulate the network delay and packet losses found in overloaded/unstable network environments.

Figure 7 shows the empirical results of the experiment. Figure 7A shows the measurement of RTT query messages, and Fig. 7B shows the time series of message arrivals and their end-to-end latency at final delivery. The pink regions mark the intervals of large-message withholding. Figure 7C shows the cumulative distribution function of the same dataset. The result shows that 80% of small messages (100 B) experienced the same or faster delivery if using the conservative policy, and in some cases the latency reduction could be about 50%. This was because the small messages were not delayed by the large messages (1 MB). The remaining 20% of small messages experienced longer delay, because of the flushing of the withheld large messages. For the large messages, about 60% of them experienced shorter latency, but the remaining 40% experienced much longer latency (Fig. 7D) due to the 5-s RTT query period. Overall, the result shows a limited view of the

**Table 3 Experiment clients configuration.**

| Client ID | Topic | QoS | Sending rate |
|---|---|---|---|
| Latency-sender | Latency | 1 | 10 KB/5 s |
| PUB1 | Msg/Small | 0 | 100 B/4 s |
| PUB2 | Msg/Large | 0 | 1 MB/4 s |
| SUB1 | Msg/#, Latency | 1 | X |
| SUB2 | Msg/# | 1 | X |
| Others 1 to 75 | Msg/# | 1 | X |

latency performance. The next subsection will give a more comprehensive analytical result and analysis.

## Analytical performance of all policies

To investigate how the proposed scheduling policies could impact the latency of large messages, we conducted analytical simulations with parameters chosen according to the empirical observations in Fig. 3. In the following, the first simulation compares different scheduling policies, and the second simulation compares different choices of RTT query period.

The simulations were conducted in a discrete-time simulator written in Python, and it took as input a time series of RTT measurements. Each latency-query message was 10 KB and each large message was 1 MB. According to the benchmark result in Fig. 3, the RTT measurement values were chosen to be randomly between 0.15 to 0.45 s, with 10% probability to have an additional latency burst duration randomly between 10 to 25 s to simulate transient network congestion. The latency burst followed an exponential distribution with $\lambda = 0.3$ to simulate the latency observed in Fig. 3. The downstream network latency for large-message (1 MB) delivery was scaled up by ten from the RTT measurements (10 KB). The end-to-end latency was the network latency plus the sojourn time in the broker. Five `threshold_l` values were simulated for comparison. The time unit was assumed to be 1 s, and a large message arrived every five time units. The simulation results spanned an interval of 5,000 time units.

### Simulation 1: scheduling policy comparison

Figure 8 shows the simulation result of large-message latency. The RTT query period was set to 1 s in this case. The starvation timer was set to 20 s. In the baseline policy for comparison (labeled as *without withholding*), there was no message withholding and each large message was sent out immediately. The gray shaded area surrounding the baseline latency gives the 95% confidence interval. The optimal latency was also shown for comparison (labeled as *clairvoyant withholding*), and the result was obtained by exhaustive search along the progress of time. For the proposed scheduling policies, each line plot shows the average latency and the 95% confidence interval at each `threshold_l` value.

As shown in Fig. 8, the conservative policy may perform better than the baseline policy if the `threshold_l` value is larger, because the broker would withhold message only when

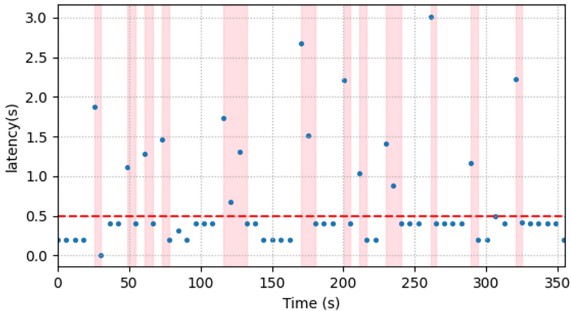

**(a)** Monitored RTT of the downstream network.

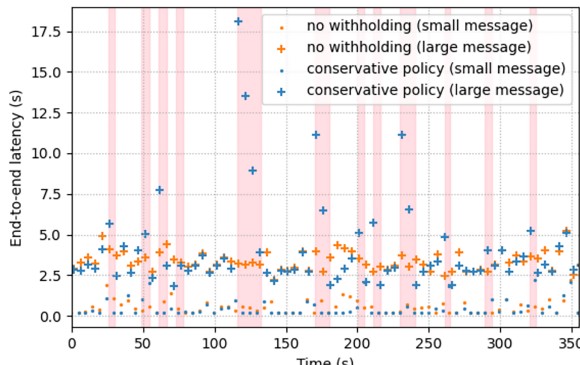

**(b)** Message arrival time and its final end-to-end latency.

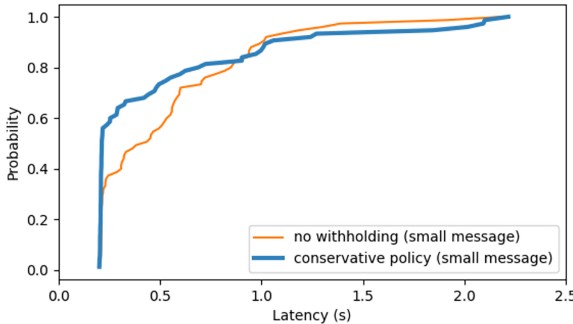

**(c)** End-to-end message latency (small message).

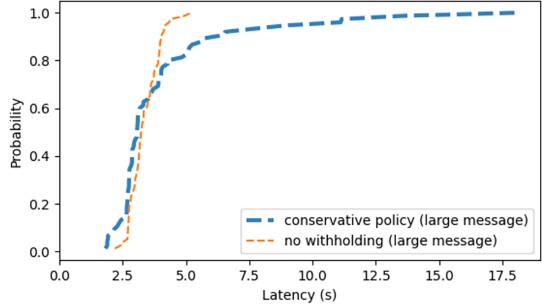

**(d)** End-to-end message latency (large message).

**Figure 7** **(A–D) Cloud experiment results.**

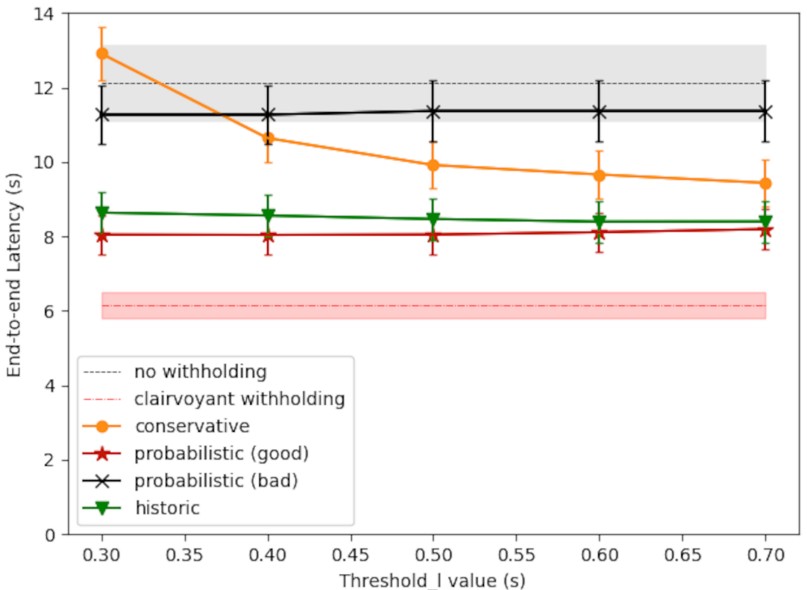

**Figure 8 Large-message latency under different scheduling policies.**

needed. The conservative policy could cause excessive latency, however, if the `threshold_l` value is too small, because in which case the withhold decision may be too sensitive to network latency fluctuations; nevertheless, the starvation timer would bound the latency to the timer value plus the one-way downstream latency.

The probabilistic policy may perform better than the conservative policy (red curve *vs.* orange curve), primarily because it takes into account $T_Q$ the waiting time before the next scheduling decision. In this way, the broker may make the right choice and send out the message right now even if $L_{now}$ is higher than `threshold_l/2`, and thereby reducing the overall latency. The latency performance of the probabilistic policy, however, depends on the accuracy of $L_{next}$, the expected network latency at the next scheduling decision. Applying Eq. (5), the 'good case' in Fig. 8 (red curve) used $\lambda = 0.057$, which is 1/17.5 and 17.5 s is the average congestion duration as set up in the generation of RTT time series. The 'bad case' in Fig. 8 (black curve) used $\lambda = 0.016$, which erroneously supposed an average congestion duration of 60 s.

In comparison, the historic policy (Eq. (6)) may provide consistent latency reduction to large messages, and its performance gain does not require any predicting of the future network condition. As shown in Fig. 8, the historic policy may save about 3.5 s in latency, comparing with the baseline policy of no message withholding. By taking into account $\Delta_S$ the message sojourn time, the historic policy has the following adaptive behavior: in the beginning of the withhold period, the broker would act conservatively in releasing the message, and thereby allowing a higher probability to pass the transient network congestion; as the withholding time extends, the broker would act more aggressively in releasing the message, unless the network congestion is very severe. Because of this, the starvation timer is not needed, and yet the message will be released at a reasonable timing.

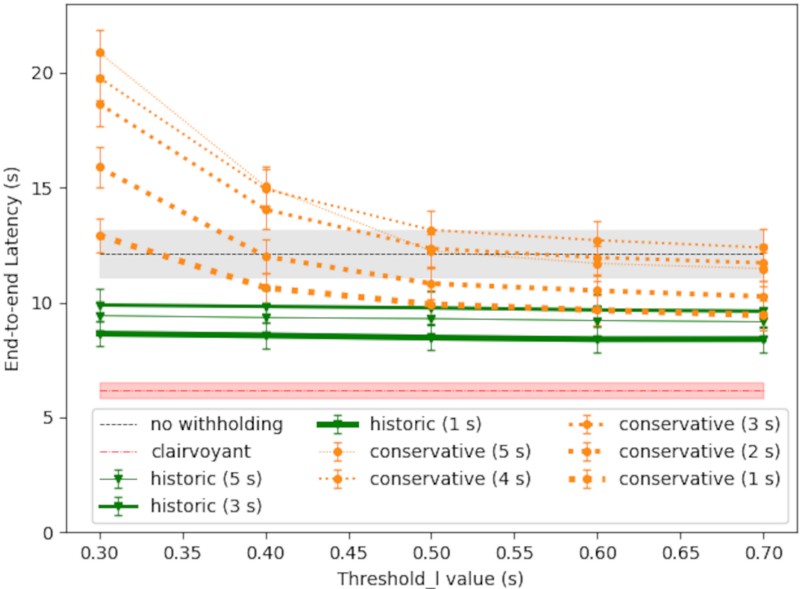

**Figure 9 Large-message latency *vs* RTT query period ($T_Q$).**

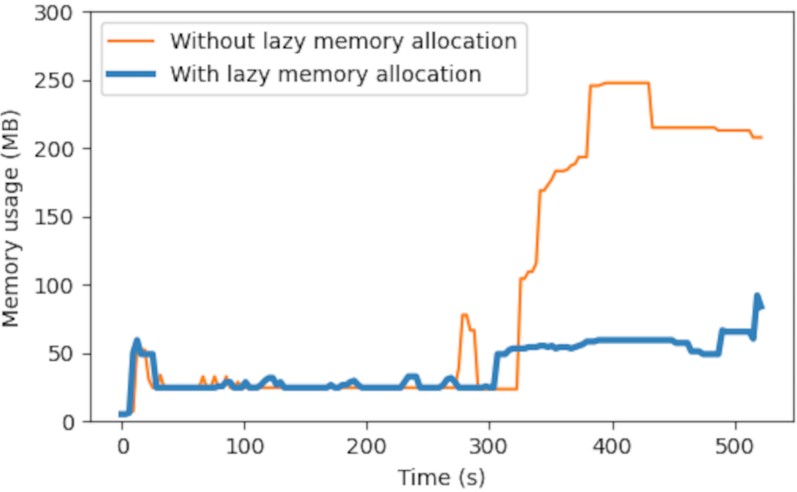

**Figure 10 Empirical memory usage in a cloud broker VM.**

### Simulation 2: impact of RTT query period

Finally, Fig. 9 shows the impact of different RTT query period ($T_Q$). In general, a shorter scheduling period could produce better latency performance, because $L_{now}$ will be based on a more recent RTT measurement. Still, there could be exceptions, for the network condition is changing along with time. For the conservative policy, with $T_Q$ set to 5 s, its latency performance was no better than the baseline policy of no withholding, and this confirms the results observed in "Empirical Performance of the Conservative Policy"; in general, with a longer $T_Q$ and a shorter `threshold_l`, the latency could approach to the starvation timer value plus the one-way downstream latency. The performance of the

historic policy could be consistently better, because the `threshold_l` value is negligible and the message sojourn time dominates Eq. (6).

### Empirical performance of broker memory footprint

Figure 10 shows the experiment result for lazy memory allocation. The experiment setup was identical to the previous one, with an addition of `tc` command applied to the outbound traffic of the broker at 250 s to simulate downstream network congestion. The result in Fig. 10 shows that the memory usage without lazy allocation then ran up to about 200 MB, which was about 20% of the total memory in the VM. This was because the queued messages accumulated and their memory allocations were not freed until final delivery. The proposed lazy memory allocation helped mitigate the problem and saved about 75% of memory usage (200 to 50 MB).

## DISCUSSION

Based on the results of performance validation, the following is a list of summary and configuration recommendations for the proposed scheduling policies:

1) The `threshold_l` value shall be large enough to reduce unnecessary message withholding.
2) The historic policy in general performs better than others, thanks to its adaption to the message sojourn time and its independence to network latency prediction.
3) Using the historic policy, the RTT query period may be set to some higher value if the monitoring overhead is of some concern.
4) With an accurate prediction or profiling of the average length of network congestion duration $\lambda$ (Eq. (5)), the probabilistic policy may perform the best among the introduced policies.

In the empirical experiments presented in "Empirical Performance of the Conservative Policy", the RTT query period $T_Q = 5$ s was chosen so that the query traffic would have a sending rate no higher than the rate of data traffic, which was 4 s (Table 3), in the hope that the RTT queries will not add too much overhead to the network environment. But as the analytical result in Fig. 9 suggests, if using the conservative policy, a reduction of $T_Q$ by seconds would reduce the end-to-end latency by the same order. Clearly, there is a trade-off, for to have a system respond quicker, the system would have to check the situation more often.

The size threshold `threshold_s` in the presented experiments was set to 10 KB because it is above most sensor data size and below most video frame size. The threshold value to be used for field applications shall be determined based on the actual size of application data. With a too-large `threshold_s`, most messages will be treated as small messages and the proposed design will not kick in; with a too-small `threshold_s`, most messages will be treated as large messages and the proposed withholding strategies might unnecessarily delay message forwarding. In either case, the lazy memory allocation still provide benefit in bounding the broker run-time memory footprint.

In the probabilistic policy, its performance depends on an estimation of the network congestion duration $\lambda$ (see Eq. (5), the good/bad cases in Fig. 8, and the presentation in "Simulation 1: Scheduling Policy Comparison"). In the case where no reliable network statistics is available, it is better to use the historic policy instead, for the latter one in making scheduling decision does not depends on $\lambda$. Further investigation could be conducted on the performance of the historic policy under different $\lambda$ values.

Finally, while the implementation in this work was based on Eclipse Mosquitto, it is noted that the proposed design may be generalized to other broker-based systems. Like what was implemented in this work, the latency detector in the proposed design (Fig. 4) can be implemented by using a publisher that is co-host with the broker. By publishing to each subscriber a designate topic known by the broker (*latency* in our experiment; see Table 3), a slightly-modified broker could perform the RTT computation upon receiving the PUBACK from each subscriber and make withholding decision.

## CONCLUDING REMARKS

This article presented a study on cloud-edge messaging and proposed improvements on the cloud MQTT broker design. By strategically withholding large-sized messages for a limited interval, the proposed solution could reduce the small-size message latency while bounding the large-size message latency. The use of lazy memory allocation effectively reduces the run-time memory footprint of the broker. The performance of the solution has been validated both empirically and analytically, and a prototype system implementation is available as an open source software (https://github.com/wangc86/Adaptive-MQTT-Transmit-Policy). In hindsight, the proposed message withhold policies seem to be general enough to be applicable to other broker-based messaging services, too. Further investigations will help to validate the feasibility.

### Funding

This research was funded by MOST grant number 109-2222-E-003-001-MY3. The funders had no role in study design, data collection and analysis, decision to publish, or preparation of the manuscript.

### Grant Disclosures

The following grant information was disclosed by the authors:
MOST: 109-2222-E-003-001-MY3.

### Competing Interests

The authors declare that they have no competing interests.

### Author Contributions

- Yi-Hsuan Tseng conceived and designed the experiments, performed the experiments, analyzed the data, performed the computation work, prepared figures and/or tables, authored or reviewed drafts of the article, and approved the final draft.

- Chao Wang conceived and designed the experiments, performed the experiments, analyzed the data, performed the computation work, prepared figures and/or tables, authored or reviewed drafts of the article, and approved the final draft.
- Yu-Tse Wei conceived and designed the experiments, performed the experiments, performed the computation work, prepared figures and/or tables, and approved the final draft.
- Yu-Ting Chiang analyzed the data, authored or reviewed drafts of the article, and approved the final draft.

## Data Availability

The raw data and code are available at Zenodo: maggie62755, & Chao Wang. (2025). wangc86/Adaptive-MQTT-Transmit-Policy: Version fo PeerJ CS article (PeerJSubmit). Zenodo. https://doi.org/10.5281/zenodo.14816709.

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
