# Peer review of "Cloud-edge MQTT messaging for latency mitigation and broker memory footprint reduction"

_PeerJ Computer Science, doi:10.7717/peerj-cs.2741_

## Round 0.1 · original submission · Minor Revisions

The reviewers found the paper interesting, and they think it can be published after some minor modifications. In particular, the authors should address the following points in the revised manuscript, in addition to all comments from the reviewers.

The related work discussion should be expanded, by considering also recently published papers. A discussion on how the experimental parameters affect the empirical results should be added. The authors should clarify how the approach generalizes to different MQTT brokers than the one considered.

The reviewers also advocated making available the tool and data, together with the instructions on how to replicate the experiments.

Reviewer 1 ·

Basic reporting

Literature references, sufficient field background/context provided. Please explain more detail (with deep analysis) about the used method, and also please explain the state of the art the used method comparing the previous methods.

Experimental design

The submission should be clearly define the research question, which must be relevant and meaningful. The knowledge gap should being investigated should be identified, and statements should be made as to how the study contributes to filling that gap.

The investigation must have been conducted rigorously and to a high technical standard. The research must have been conducted in conformity with the prevailing ethical standards in the field.

Validity of the findings

Decisions are not made based on any subjective determination of impact, degree of advance, novelty or being of interest to only a niche audience. We will also consider studies with null findings. Replication studies will be considered provided the rationale for the replication, and how it adds value to the literature, is clearly described. Please note that studies that are redundant or derivative of existing work will not be considered.

Additional comments

1. Formal results still not included clear definitions of all terms and theorems, and detailed proofs.
2. Methods should be described with sufficient information to be reproducible by another investigator.

Reviewer 2 ·

Basic reporting

The manuscript titled "Cloud-Edge MQTT Messaging for Latency Mitigation and Broker Memory Footprint Reduction" explores challenges in cloud-edge communication, particularly focusing on MQTT, a widely used IoT messaging protocol. The study identifies two primary issues with MQTT brokers in smart-city applications:

Latency Issues: Transient network congestion disproportionately affects large messages, causing delays for smaller, latency-sensitive messages.
Memory Footprint: The accumulation of large undelivered message copies in the broker leads to excessive memory usage, which is particularly problematic in constrained environments like cloud VMs.
The English is mostly clear, but there are several instances of grammatical errors that need correction.
Example:
In the abstract: "Empirical and analytical validations has demonstrated..." should be "Empirical and analytical validations have demonstrated...".
In Section 3: "QoS 2, applications typically request either QoS 0 or QoS 1, and this article also focuses on the two QoS levels." could be revised for clarity as "QoS 2 is rarely used; hence, this study focuses on QoS 0 and QoS 1."
The related work section provides a strong foundation by referencing notable studies on MQTT and cloud-edge messaging. However, the authors should consider including more recent articles (published within the last 2–3 years) to ensure the manuscript reflects the latest advancements in MQTT research.

Experimental design

The manuscript describes the experimental setup in sufficient detail; however, the justification for specific parameter choices, such as the RTT query period and message size thresholds, is not adequately discussed.
While the study evaluates multiple scheduling policies, it does not comprehensively explore the sensitivity of results to variations in parameters like message size thresholds (threshold s) and network congestion duration (λ).
The experiments focus on Eclipse Mosquitto as the MQTT broker implementation. It is recommended to discuss whether the findings and proposed solutions are generalizable to other MQTT broker implementations or IoT messaging protocols. (EMQx, HiveMQ, Fluux etc..)
The manuscript includes a detailed description of the experimental design, such as the use of AWS EC2 instances and Linux tools for traffic simulation. However, it would be beneficial to include a step-by-step methodology or code repository link to facilitate replication by other researchers.

Validity of the findings

Suggestions to enhance impact:
Compare the findings with state-of-the-art MQTT optimization approaches quantitatively to further emphasize the novelty.
The proposed solutions, though designed for MQTT, can likely extend to other broker-based protocols or messaging systems.

Reviewer 3 ·

Basic reporting

See report

Experimental design

See report

Validity of the findings

See report

Additional comments

Overall, the paper is good, organised, well written and the content is significant for the field related to the problem-based learning approaches in computer science.
Some strengths related to both scientific content and form are:
- Based on previous similar works, the authors identified the unsolved challenges and proposed a study to examine how they could less influence the problem-based learning process.

Some weaknesses/suggestions are:
- The Abstract should emphasize better the fact that the present study is a continuation of a previous research of the other authors.
- The authors should emphasize better the originality of the paper related to previous studies.
- The reviewer fails to see the research related to specified domain. Also missing the comparative analysis as to existing approaches.

---

## Round 0.2 · accepted · Accept

I carefully read the revision and the authors' response to reviewers. The authors addressed all reviewers' comments and improved the overall presentation of the paper. The paper is ready for publication.